# Advanced Control Method of 5-Phase Dual Concentrated Winding PMSM for Inverter Integrated In-Wheel Motor

Kan Akatsu [1,*] and Keita Fukuda [2,*]

1   Department of Mathematics, Physics, Electrical Engineering and Computer Science, Yokohama National University, Kanagawa 240-8501, Japan
2   Department of Electrical Engineering and Computer Science, Shibaura Institute of Technology, Tokyo 135-8548, Japan
*   Correspondence: akatsu-kan-py@ynu.ac.jp (K.A.); ma18088@shibaura-it.ac.jp (K.F.)

**Abstract:** This paper presents some techniques for driving novel 5 phase dual winding PMSM (Permanent Magnet Synchronous Motor) for the in-wheel motor. The motor realizes winding change over characteristics that can expand driving area from high-torque mode to high-speed mode due to the dual winding construction. However, the dual winding structure makes a high-current ripple due to high coupling between windings. The paper proposes some control methods to reduce the current ripple, including inverter career ripple. The paper also presents harmonics current injection, such as the 3rd harmonics current injection method, to reduce the torque ripple and generate higher torque.

**Keywords:** permanent magnet synchronous motor; dual windings structure; 5-phase driving; winding changeover; disturbance observer





## 1. Introduction

An integrated permanent magnet synchronous motor (PMSM) coupled with an inverter is part of a mainstream effort to develop a small traction system for electric vehicles (EVs) [1]. To make the power train system more compact, the use of a wide band gap power device, such as a SiC (Silicon Carbide) MOS-FET, and a GaN (Gallium Nitride) device, is an attractive prospect for the loss reduction of the inverter. This is because the cooling requirement becomes simpler for the integration system. However, the current density of these chips is lower than that of the Si device chips, such as IGBT (Insulated Gate Bipolar Transistor). Additionally, with the use of a high-current capacity wide band gap module, several parallel connections of the chips are required, the size of the inverter increases, and the integration becomes more challenging.

An ultimate integration system for EVs is the in-wheel motor (IWM), which has been under development for the past few decades [2,3]. One of the challenges in realizing IWM is the efficient cooling of the system. Similar to how the power cable connected to the wheel is not preferred by car manufacturers, a water-cooling hose connected to the wheel is not preferred for EVs, as it results in being bulky and failure-prone. Therefore, a simpler air-cooling system is desired. Additionally, a fault-tolerant system must be equipped for the IWM so as not to stop the EV when the motor or inverter fails [4].

With the use of wide-band gap devices for the integrated IWM, an air-cooling system can be realized, making the IWM system more achievable by introducing a fault-tolerant motor system. This will make EVs a more attractive prospect in the automobile industry, contributing to the reduction in $CO_2$ from gasoline-powered cars.

To realize a fault-tolerant air-cooled IWM, this study proposes a 5-phase dual winding PMSM driven by ultra-small SiC modules. The entire power and mechanical gear systems are integrated into a 16-inch wheel, which can be attached to the existing Nissan LEAF without any additional modifications. The authors have already presented the system

construction, motor structure, inverter structure, and mechanical gears along with the experimental results [5]. This paper presents some techniques for driving novel 5-phase dual winding PMSM for the in-wheel motor. The motor realizes winding change over characteristics [6,7] that can expand driving area from high-torque mode to high-speed mode due to the dual winding construction. However, the dual winding structure makes a high-current ripple due to high coupling between windings. The paper proposes some control methods to reduce the current ripple, including inverter career ripple. The paper also presents harmonics current injection, such as third harmonics current injection method, to reduce the torque ripple and generate higher torque.

In the paper, the proposed in-wheel motor system is described first, the motor design and winding change over technique by the proposed motor are explained in Section 3. Further, the current control method for the dual winding 5-phase machine is proposed in Section 4, some experimental results to verify the method are shown in Section 5.

## 2. Target In-Wheel Motor System

The proposed in-wheel motor system was designed to replace the Nissan Leaf™ 1st generation wheel. Since the motor output for the vehicle is 80 kW, the maximum output of one wheel is 40 kW, and it is set to 20 kW at a continuous rating. The torque and rotational speed were determined as shown in Table 1, considering the reduction gear ratio (about 1.8) from the torque required for driving and the maximum speed.

**Table 1.** Motor specifications.

| Parameter | Value |
| --- | --- |
| Torque [Nm] | 72 |
| Max. speed [rpm] | 20,000 |
| DC bus voltage [V] | 360 |
| Current [Arms] (peak) | 55 (78) |
| Max. Output power [kW] | 40 |
| Motor diameter [mm] | 159 |
| Motor length [mm] | 82 |

In the in-wheel motor, it is desirable that the wiring connected to the motor is as little as possible. This is because vibration during running may induce a break, and it is also to avoid wiring freezing in snow running. Therefore, in this system, the inverter was made into a machine and electric structure integrated into the motor, and the high-power electrical wiring to the wheel was made to be DC wiring only. In addition, it was designed with the aim of natural air cooling by running wind in addition to the water cooling hose [8].

In addition, a multi-phase motor is assumed to realize a fail-safe system that prevents the event of inverter failure. As mentioned above, the inverter of this system uses SiC so that it can withstand high temperatures because it assumes natural air cooling. Since the current density of the SiC chip is still smaller than that of IGBT, multi-phase structure is also effective in reducing the current per phase. In this research, an ultra-small SiC half-bridge module, which has been developed by authors [9], is used to make a highly integrated in-wheel motor. Figure 1 shows a picture of the developed in-wheel motor and its CAD model, Figure 2 shows a picture of the SiC half-bridge module.

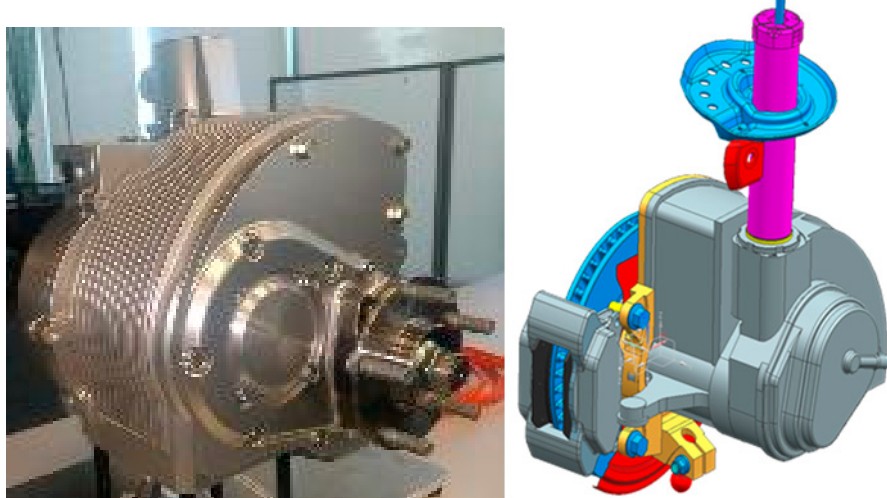

**Figure 1.** Proposed in-wheel motor system. The system includes DC capacitor, inverter, controller, and reduction gears in the unit. Natural air cooling is used to continuously output 20 kW drive.

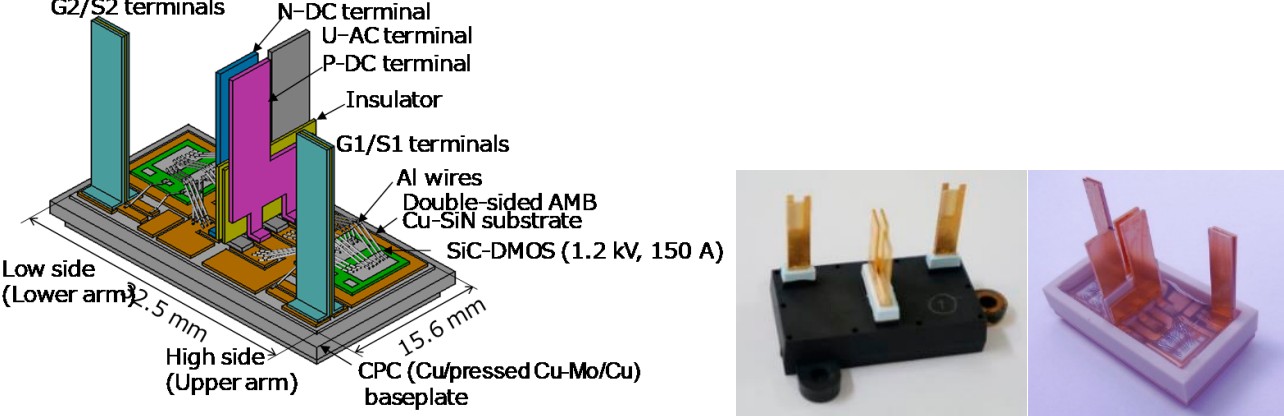

**Figure 2.** Developed ultra-small SiC half-bridge inverter module. 1200 V-150 A rated and the module was confirmed to work under 250 degrees temperature conditions.

The details of the motor design are described in the next section.

## 3. Motor Design

### 3.1. Phase Number and Pole Number Selection

As mentioned in Section 2, a fail-safe function is required for in-wheel motors. Therefore, it was chosen to drive with a phase number larger than the conventional three phases. In addition, the concentrated winding was selected in order to reduce the motor coil end length as much as possible. Figure 3 shows the relationship between phase numbers and winding factors. From Figure 3, it can be seen that the winding factor of odd phase numbers is high. In this motor, five phases [10], which are the smallest phase numbers of three phases or more, were selected in the odd phase number. This is to avoid the inverter module increase due to the increase in the number of phases. As a result, the 8 pole 10 slot 5-phase structure was set to a winding coefficient of 0.95 in a fundamental flux linkage [11–13]. Figure 4 shows a half-model of the motor structure and a picture of the rotor and stator. Nd-Fe-B magnet with a V-shaped structure that can increase the number of magnetic flux linkage was used.

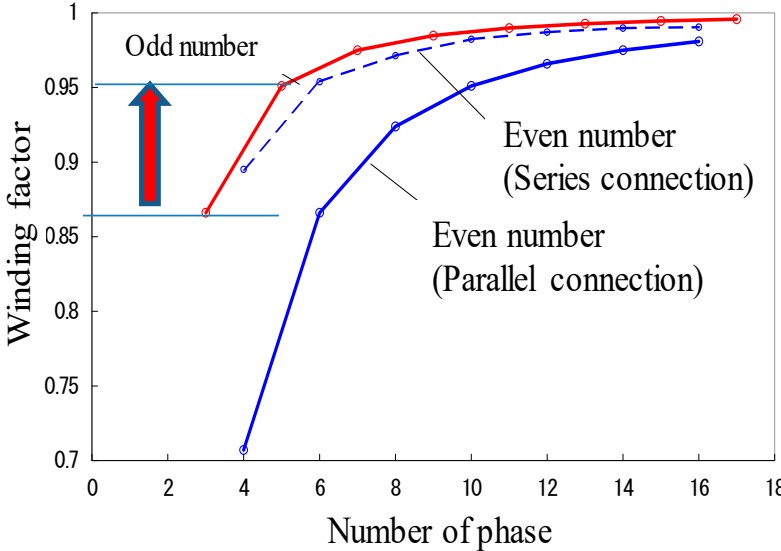

**Figure 3.** Winding factor of multi-phase machine. 5-phase concentrated winding has a winding factor of 0.95.

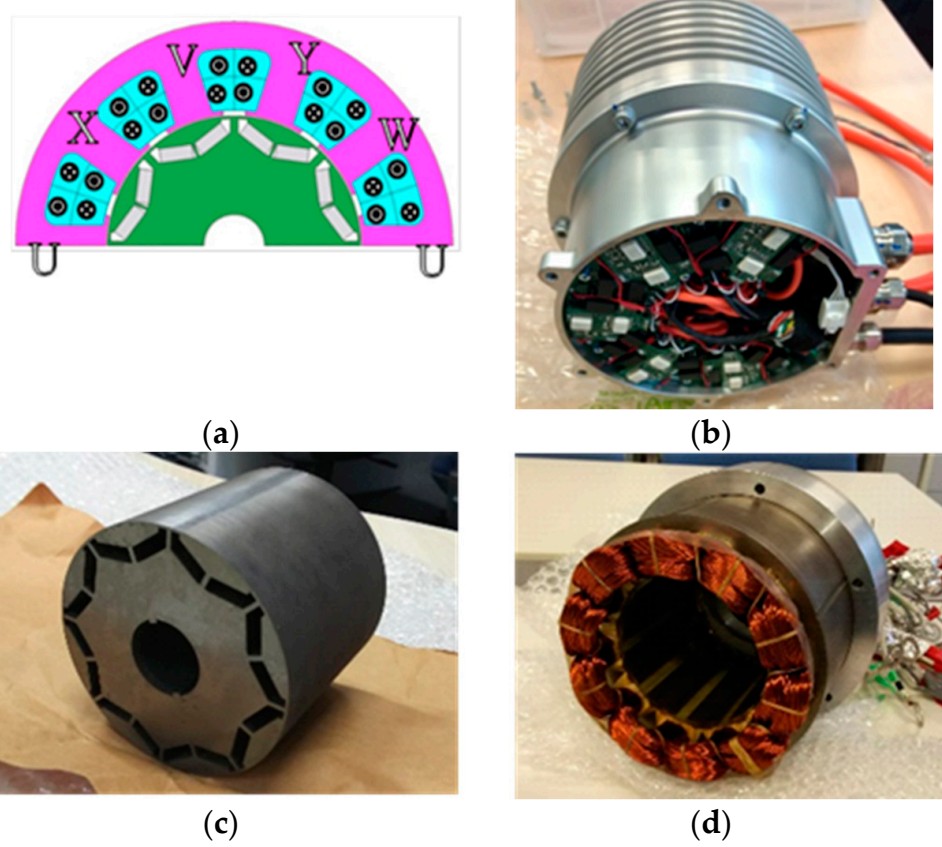

**Figure 4.** 10 slots 8 poles 5-phase motor model. The rotor was designed to ensure high-speed rotation up to 20,000 rpm. 0.35 mm electrical steel sheet is used and maximum stress at 20,000 rpm is around 100 MPa at the edge of the magnet in the rotor. (**a**) FEA (Finite Element Analysis) 1/2 model of the motor, (**b**) inverter integrated assembled motor, (**c**) rotor without magnet assembly, (**d**) stator with dual windings.

### 3.2. Winding Change over Technique

Once the number of turns was calculated to satisfy the specifications shown in Table 1, it was 28 turns, and the output power was not achieved in the medium-speed range. Moreover, the efficiency decreases due to flux weakening in the high-speed range were remarkable. Then, this motor used a winding change over technique [6,14] by dual winding in a stator tooth as a fail-safe system. Figure 5 shows an inverter circuit. There are two systems that drive each winding of the five phases with a half-bridge inverter, and two windings are wound in one stator tooth of the motor. By calculating the output characteristics with changing the number of turns in each winding, a combination of 17 turns and 11 turns for windings was selected [15]. As shown in Figures 4a and 5, one stator tooth has two windings, an outer winding and an inner winding. These windings are electrically isolated, the flux linkages are shared. Thus, the induced voltage to the inverter is decided by a higher number of turn winding, outer winding because the windings are parallel connected to the inverter.

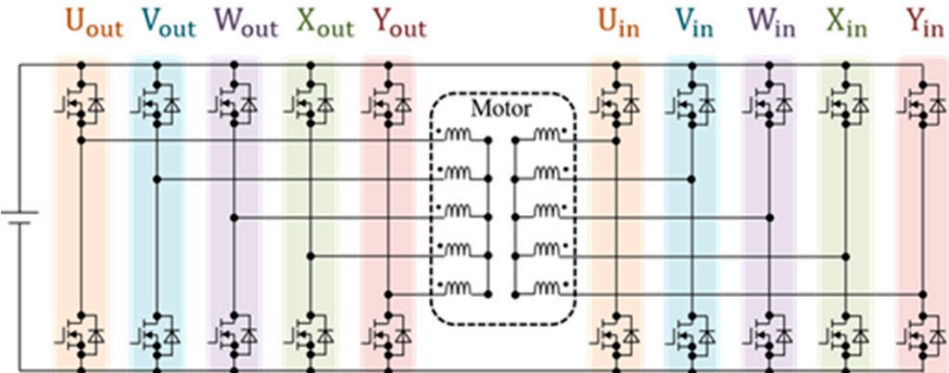

**Figure 5.** Inverter structure for 5-phase motor. SiC half-bridge module in Figure 2 is used for each half-bridge.

In addition, three types of output characteristics shown in Figure 6 can be realized by selecting the winding. Figure 7a shows the output characteristics in each mode A, B, and C. The high-speed mode C can drive 20,000 rpm without flux weakening current. As a study of the number of turn selections, Figure 7b,c shows the output characteristics when the combination of the number of turns is 23-5 and 14-14, respectively. In the design, the maximum output power of 40 kW is obtained only by 17-11 combination.

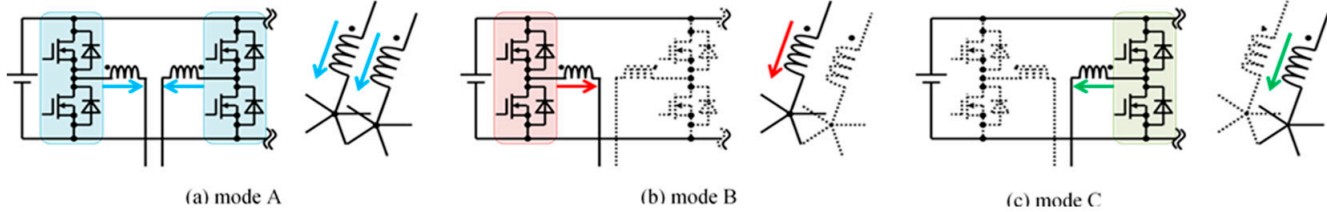

**Figure 6.** Possible driving methods. (**a**) Mode A is a high-torque mode in which both windings are used, however, the EMF is smaller than the conventional 28 turn structure. (**b**) Mode B uses 17 turn winding. (**c**) Mode C uses 11 winding in the high-speed region.

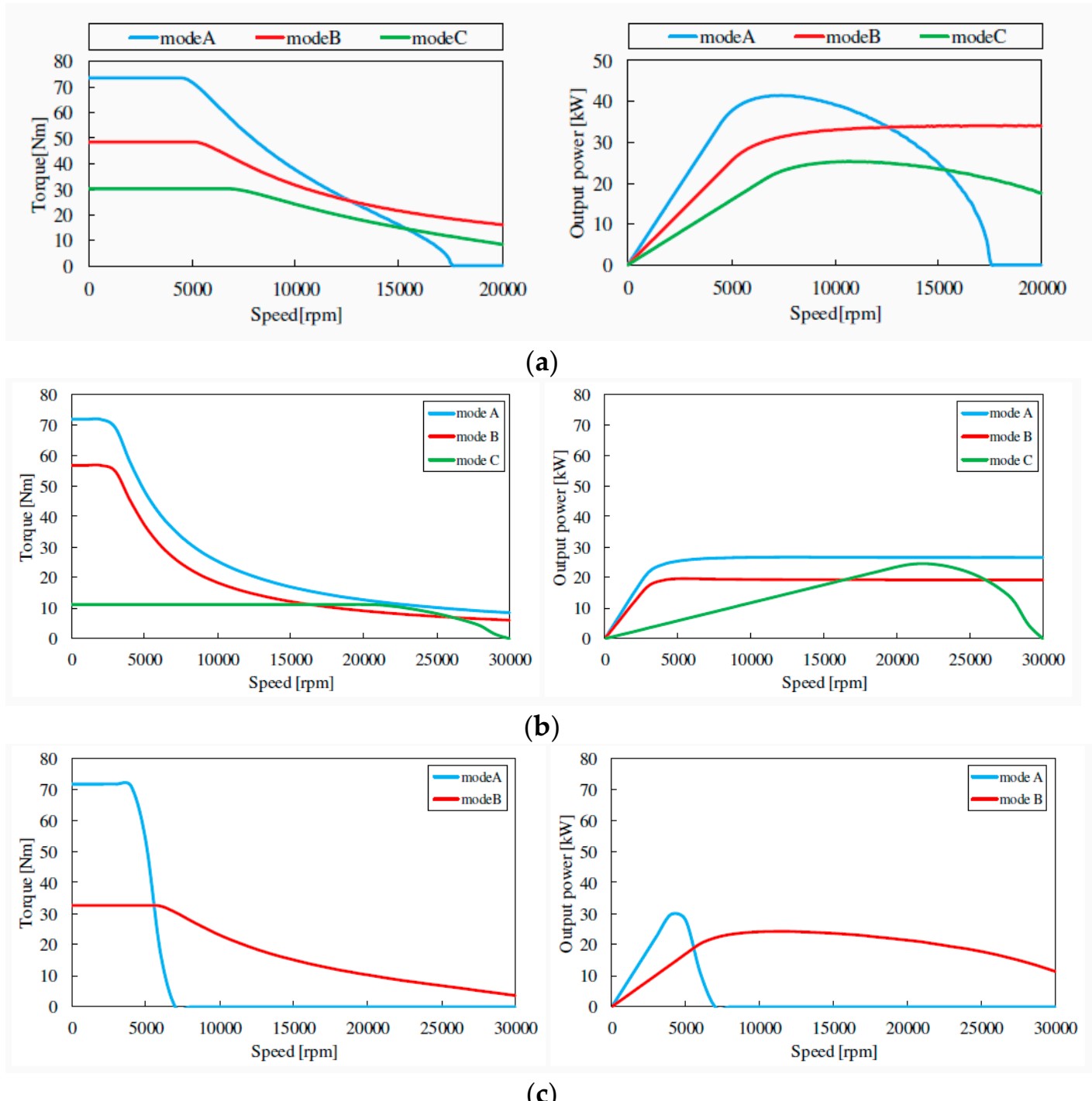

**Figure 7.** (**a**) Output characteristics of the proposed 3 mode motor. (Left) Torque-Speed curve, (Right) Output power-Speed curve. (**b**) Output characteristics of the 25 turn-3 turn motor. (Left) Torque-Speed curve, (Right) Output power-Speed curve. (**c**) Output characteristics of the 14 turn-14 turn motor. (Left) Torque-Speed curve, (Right) Output power-Speed curve.

As shown in Figure 8, it is possible to improve the efficiency by 3% for normal driving motor with flux weakening current. In this drive, since it is possible to switch windings without suddenly releasing the energy stored in the coil at the time of winding switching, it is characterized by almost no torque shock at the time of switching.

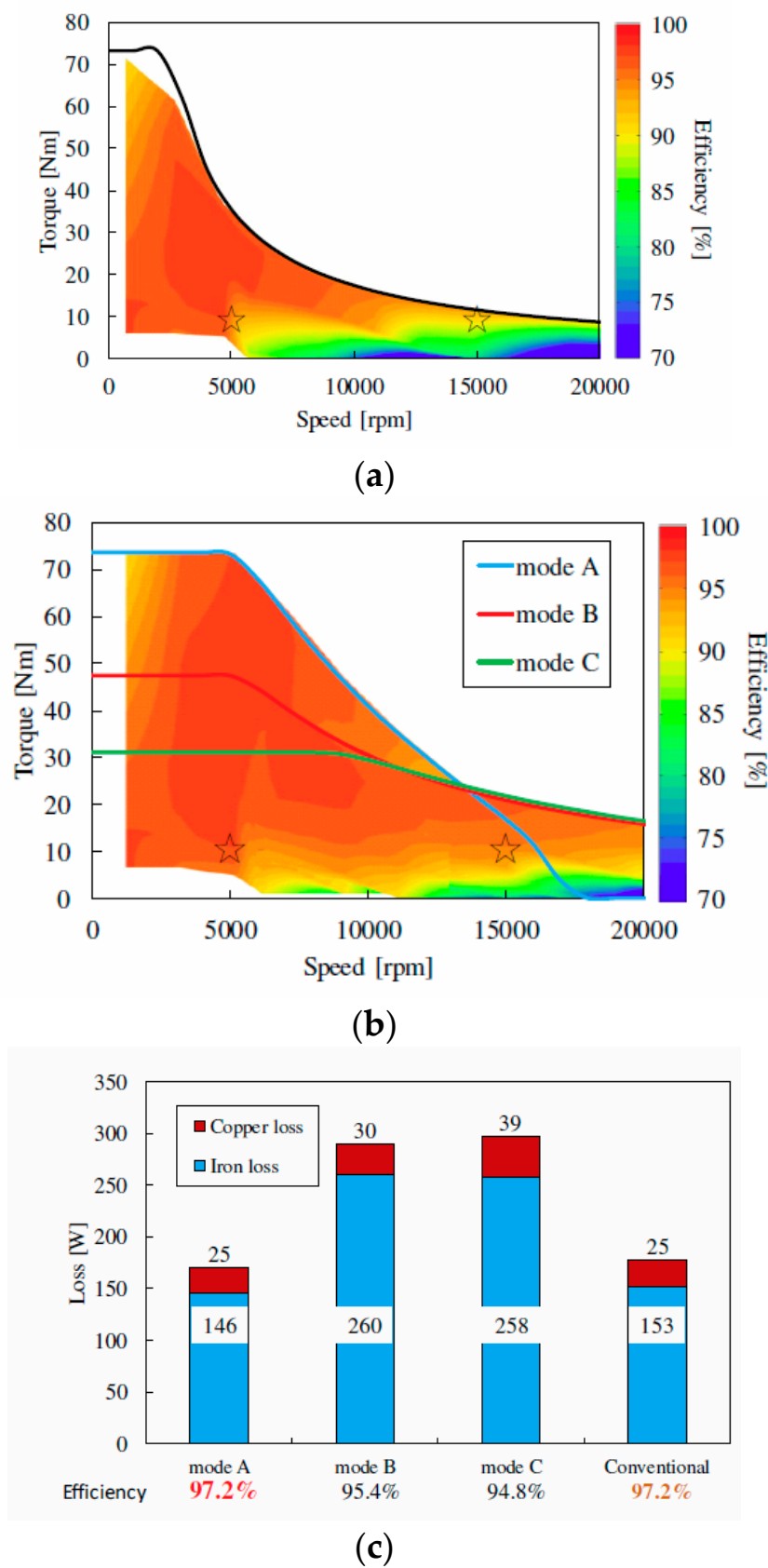

**Figure 8.** *Cont.*

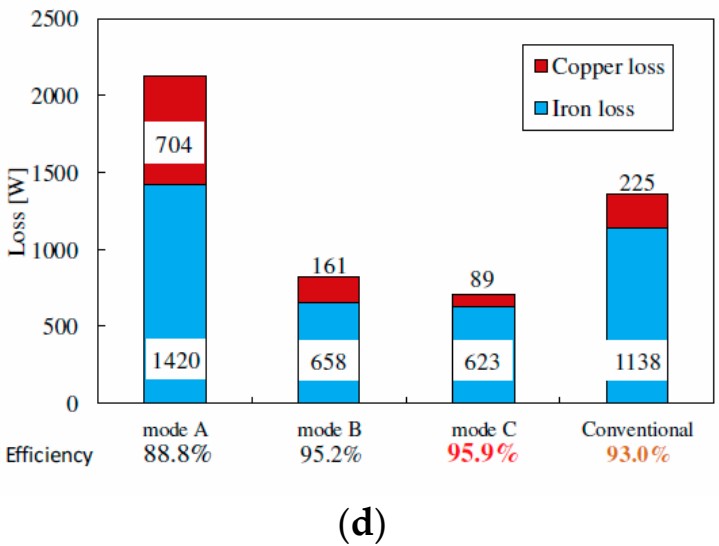

**(d)**

**Figure 8.** Calculated efficiency map in the conventional and proposed motor. Mode C can improve the efficiency by around 3% in the high-speed region. (**a**) 28 turn conventional motor, (**b**) 17-11 turn 3 mode motor, (**c**) efficiency comparison at 10 Nm 5000 rpm in each mode, (**d**) efficiency comparison at 10 Nm 15,000 rpm in each mode.

## 4. Motor Control Design

Since this motor is a 5-phase motor, a third harmonic current that does not appear in a conventional 3-phase motor should be controlled [16]. In addition, if a coupling between inner and outer windings appears, a decoupling control between dual windings is necessary. Due to the mutual inductance, the inverter carrier ripple increases. In this section, these solutions by motor control are described.

### 4.1. Current Control

Equation (1) shows a voltage differential equation synchronized on fundamental frequency, Equation (2) shows it synchronized on 3rd harmonics frequency [17,18]. In these equations, **L** is a self-inductance and **M** is mutual inductance between windings. The suffix out and in indicates the outer winding and the inner winding, respectively. For control of the fundamental frequency current, the inductance $L_{dq}$ is much larger than the 3rd harmonics inductance $L_{3dq}$, the induced voltage by the 3rd order inductance can be ignored. However, due to the lower inductance, the PWM voltage generates high harmonics current. In this research, the fundamental current and the 3rd harmonics current are controlled by using a disturbance observer that can compensate mutual inductance effect, and PWM phase shift control is used to decrease the high harmonics current.

$$\begin{bmatrix} v_{dqout} \\ v_{dqin} \end{bmatrix} = \begin{bmatrix} R_{out} & 0 \\ 0 & R_{in} \end{bmatrix}\begin{bmatrix} i_{dqout} \\ i_{dqin} \end{bmatrix} + \begin{bmatrix} sL_{dqout} & sM_{dq} \\ sM_{dq} & sL_{dqin} \end{bmatrix}\begin{bmatrix} i_{dqout} \\ i_{dqin} \end{bmatrix} + \omega\begin{bmatrix} JL_{dqout} & JM_{dq} \\ JM_{dq} & JL_{dqin} \end{bmatrix}\begin{bmatrix} i_{3dqout} \\ i_{3dqin} \end{bmatrix} + \omega\begin{bmatrix} \psi_{dqout} \\ \psi_{dqin} \end{bmatrix} \quad (1)$$

$$\begin{bmatrix} v_{3dqout} \\ v_{3dqin} \end{bmatrix} = \begin{bmatrix} R_{out} & 0 \\ 0 & R_{in} \end{bmatrix}\begin{bmatrix} i_{3dqout} \\ i_{3dqin} \end{bmatrix} + \begin{bmatrix} sL_{3dqout} & sM_{3dq} \\ sM_{3dq} & sL_{3dqin} \end{bmatrix}\begin{bmatrix} i_{3dqout} \\ i_{3dqin} \end{bmatrix} + 3\omega\begin{bmatrix} JL_{3dqout} & JM_{3dq} \\ JM_{3dq} & JL_{3dqin} \end{bmatrix}\begin{bmatrix} i_{3dqout} \\ i_{3dqin} \end{bmatrix} + 3\omega\begin{bmatrix} \psi_{3dqout} \\ \psi_{3dqin} \end{bmatrix}$$
$$\approx \begin{bmatrix} R_{out} & 0 \\ 0 & R_{in} \end{bmatrix}\begin{bmatrix} i_{3dqout} \\ i_{3dqin} \end{bmatrix} + 3\omega\begin{bmatrix} \psi_{3dqout} \\ \psi_{3dqin} \end{bmatrix} \quad (2)$$

$$\boldsymbol{v}_{dq} = \begin{bmatrix} v_d & v_q \end{bmatrix}^{\mathrm{T}}, \quad \boldsymbol{v}_{3dq} = \begin{bmatrix} v_{3d} & v_{3q} \end{bmatrix}^{\mathrm{T}}$$

$$\boldsymbol{i}_{dq} = \begin{bmatrix} i_d & i_q \end{bmatrix}^{\mathrm{T}}, \quad \boldsymbol{i}_{3dq} = \begin{bmatrix} i_{3d} & i_{3q} \end{bmatrix}^{\mathrm{T}}$$

$$\boldsymbol{\psi}_{dq} = \begin{bmatrix} \psi_d & \psi_q \end{bmatrix}^{\mathrm{T}}, \quad \boldsymbol{\psi}_{3dq} = \begin{bmatrix} \psi_{3d} & \psi_{3q} \end{bmatrix}^{\mathrm{T}}$$

$$\boldsymbol{L}_{dq} = \begin{bmatrix} L_d & 0 \\ 0 & L_q \end{bmatrix}, \quad \boldsymbol{L}_{3dq} = \begin{bmatrix} L_{3d} & 0 \\ 0 & L_{3q} \end{bmatrix} \tag{3}$$

$$\boldsymbol{M}_{dq} = \begin{bmatrix} M_d & 0 \\ 0 & M_q \end{bmatrix}, \quad \boldsymbol{M}_{3dq} = \begin{bmatrix} M_{3d} & 0 \\ 0 & M_{3q} \end{bmatrix}$$

$$\boldsymbol{J} = \begin{bmatrix} 0 & -1 \\ 1 & 0 \end{bmatrix}$$

From Equation (1), the inductances can be obtained by the following equations and the output torque is calculated as Equation (5).

$$L_{dout} = \frac{N_{out}\left(v_{qout} - R_{out}i_{qout} - \omega\psi_{out}\right)}{\omega\left(N_{out}i_{dout} + N_{in}i_{din}\right)}, \quad L_{din} = \frac{N_{in}\left(v_{qin} - R_{in}i_{qin} - \omega\psi_{in}\right)}{\omega\left(N_{out}i_{dout} + N_{in}i_{din}\right)}$$

$$L_{qout} = \frac{N_{out}\left(R_{out}i_{dout} - v_{dout}\right)}{\omega\left(N_{out}i_{qout} + N_{in}i_{qin}\right)}, \quad L_{qin} = \frac{N_{in}\left(R_{in}i_{din} - v_{din}\right)}{\omega\left(N_{out}i_{qout} + N_{in}i_{qin}\right)}$$

$$M_d = \frac{N_{out}\left(v_{qin} - R_{in}i_{qin} - \omega\psi_{in}\right)}{\omega\left(N_{out}i_{dout} + N_{in}i_{din}\right)} = \frac{N_{in}\left(v_{qout} - R_{out}i_{qout} - \omega\psi_{out}\right)}{\omega\left(N_{out}i_{dout} + N_{in}i_{din}\right)} \tag{4}$$

$$M_q = \frac{N_{out}\left(R_{in}i_{din} - v_{din}\right)}{\omega\left(N_{out}i_{qout} + N_{in}i_{qin}\right)} = \frac{N_{in}\left(R_{out}i_{dout} - v_{dout}\right)}{\omega\left(N_{out}i_{qout} + N_{in}i_{qin}\right)}$$

$$T = P\{\psi_{dout}i_{qout} + \psi_{din}i_{qin} + \left(L_{dout} - L_{qout}\right)i_{dout}i_{qout} + \left(L_{din} - L_{qin}\right)i_{din}i_{qin} + \left(M_d - M_q\right)\left(i_{dout}i_{qin} + i_{din}i_{qout}\right)\} \tag{5}$$

Figure 9a–c shows the analyzed results of inductances. Figure 9a shows the inductances at mode A that uses both inner and outer windings. Figure 9b,c shows the inductances at mode B and mode C, respectively. Only the outer winding is used in mode B, and mode C uses only inner winding. Figure 10 shows calculated torque by Equation (5) and analyzed torque by FEA (Finite Element Analysis). The calculated torque is consistent with the FEA results.

Figure 11 shows a block diagram of current control. Fundamental frequency current is controlled by using a disturbance observer that can compensate for effects from the mutual inductances between windings. Third harmonics current control is independently designed with the fundamental one, compensated voltage to control 3rd harmonics current is added to the fundamental voltage reference.

### 4.2. 3rd Harmonics Current Control

One of the features of the 5-phase motor is that it is possible to use the 3rd harmonics current. Since the IPM motor has a 3rd harmonic of the magnet flux, torque can be improved by injecting the 3rd harmonic current [19]. Figure 12 shows the torque amplitude change phenomenon when the 3rd harmonics current is added to the fundamental current. The amplitude of the fundamental current is 30 A, and maximum 15 A 3rd harmonics current is added. It is possible to improve the torque by more than 20%. Moreover, the 3rd harmonics current can reduce the torque ripple by adjusting the phase of the 3rd harmonics current. Figure 13 shows the torque ripple ratio when the 3rd harmonics current phase and amplitude are adjusted. The waveforms of the output torque are shown in Figure 13b. It is possible to reduce the torque ripple by around 50% by adjusting the phase and amplitude of the current.

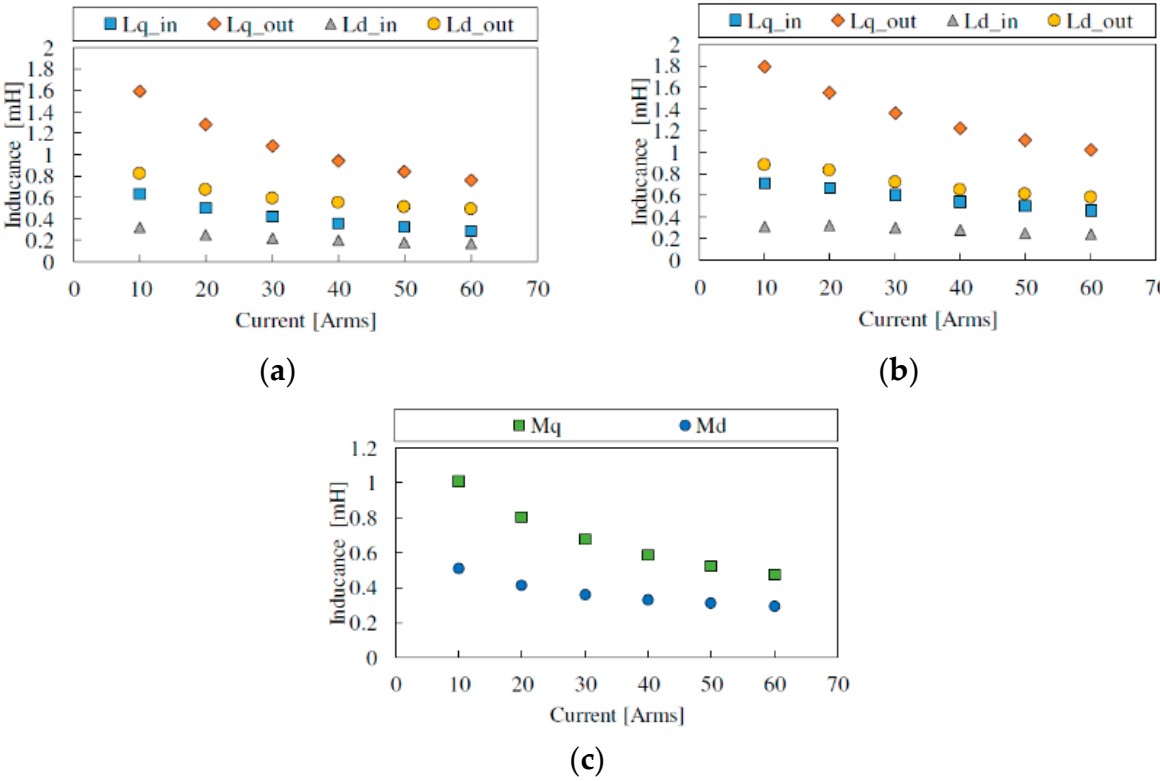

**Figure 9.** (**a**) Inductances at mode A, (**b**) Inductances at mode B, C, (**c**) Mutual inductances.

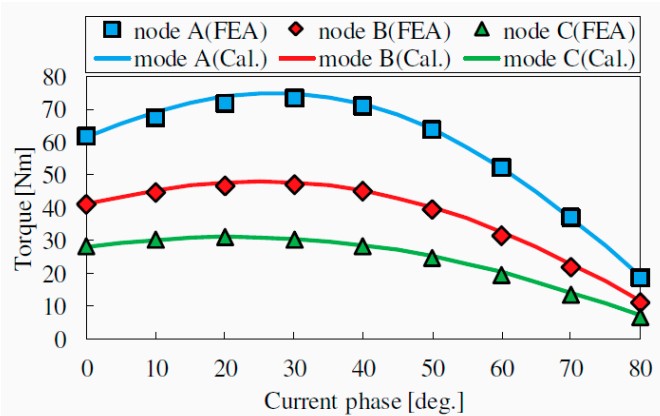

**Figure 10.** Calculated torque by Equation (5) and analyzed torque by FEA.

### 4.3. PWM Phase Shift Control

Although the induced voltage can be ignored in the 3rd harmonics coordinate system in Equation (2), a step voltage such as PWM creates a large ripple current. Because of their high frequency, it is difficult to remove by the disturbance observer shown in Figure 11. In this study, the carrier harmonic current is suppressed by shifting the phase of the PWM carrier by using the relatively close number of windings between the inner and outer windings. Figure 14 shows the phase shift carrier waveforms, the inner winding carrier has 180 degree phase shift with the outer winding carrier. Figure 15 shows the phase current waveforms with/without carrier phase shift and their FFT (First Fourier Transform) results. The carrier frequency is 20 kHz and fundamental frequency is 100 Hz. It is clear that the PWM carrier ripple is decreased by the carrier phase shift. It is noted that there is a drawback to DC current from the battery that DC current ripple is increased. Figure 16 shows DC current ripple. Then, the capacitance of DC capacitor should be carefully chosen.

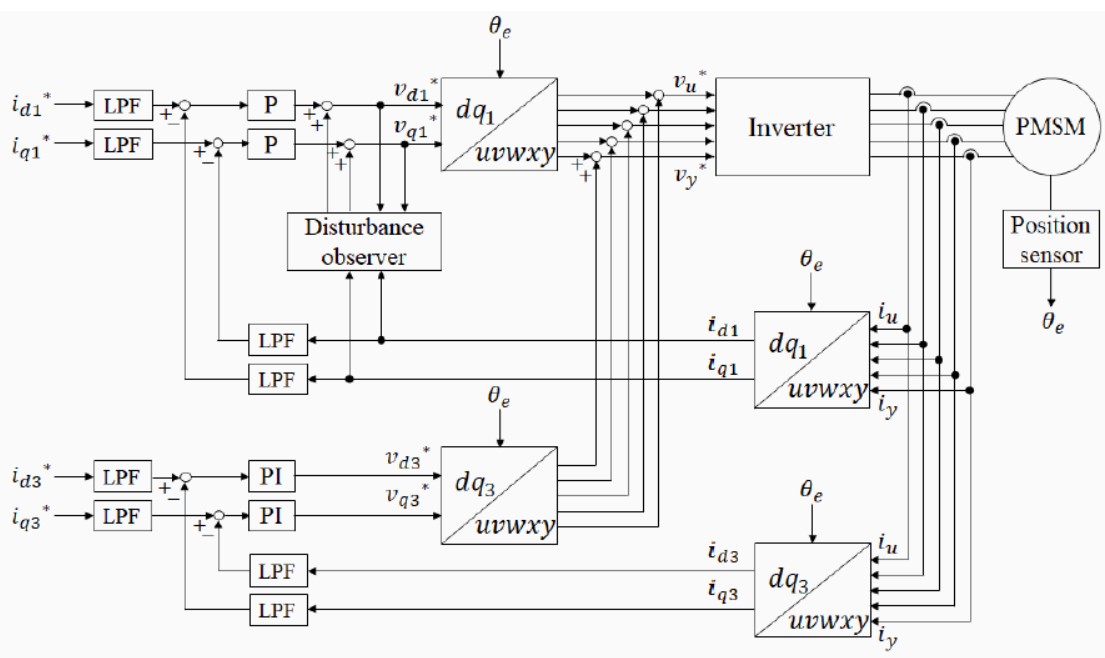

**Figure 11.** Block diagram of current control.

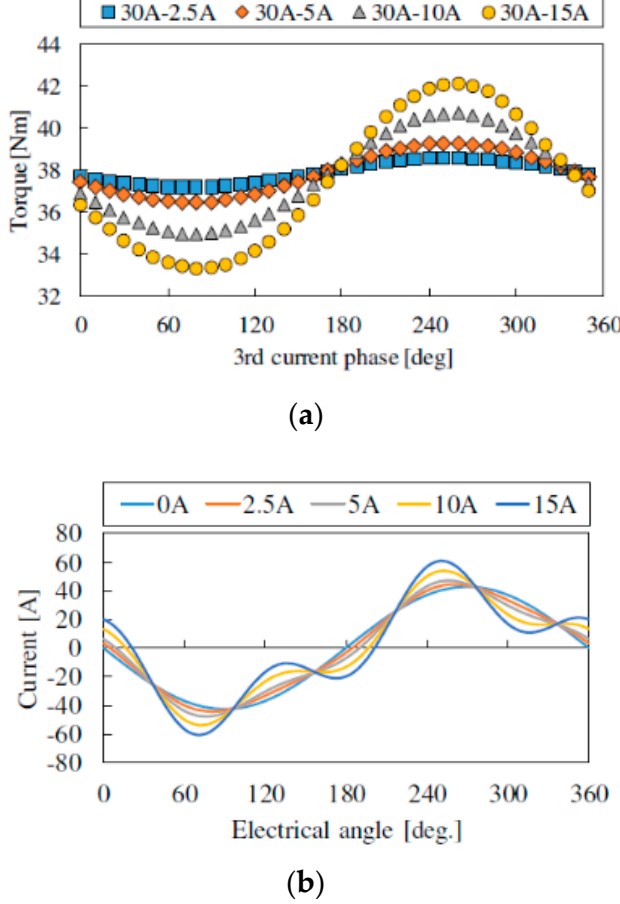

**(a)**

**(b)**

**Figure 12.** (**a**) Output torque amplitude with 3rd harmonics current. Fundamental current is 30 A, 2.5 A–15 A harmonics are added. (**b**) Current waveforms when 3rd harmonics is added to the fundamental current.

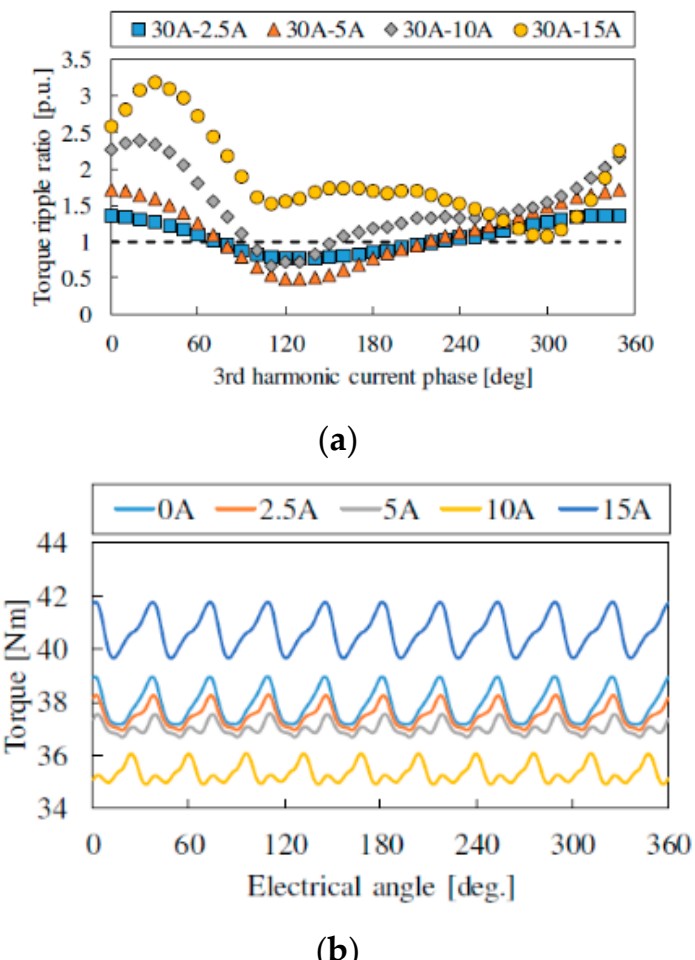

(a)

(b)

**Figure 13.** (**a**) Torque ripple ratio characteristics with 3rd harmonics current. Fundamental current is 30 A, 2.5 A–15 A harmonics are added. (**b**) Output torque waveforms when 3rd harmonics is added to the fundamental current.

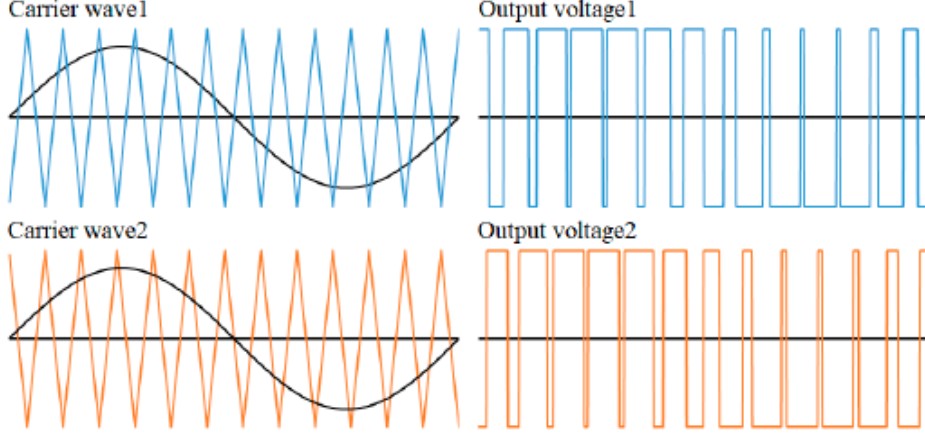

**Figure 14.** Carrier phase shift control.

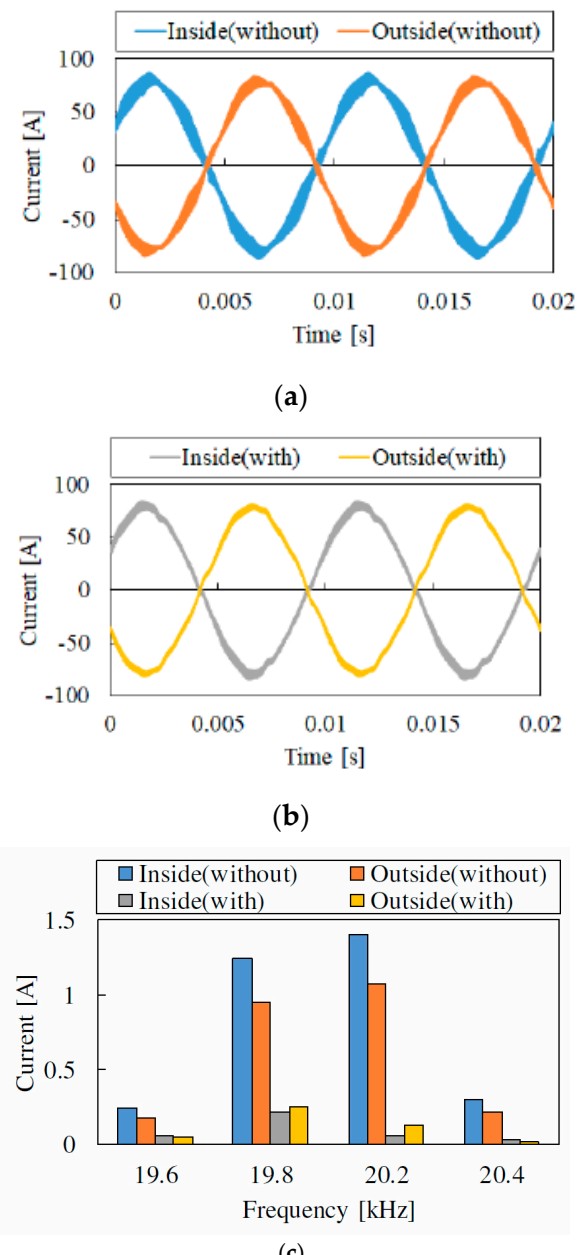

**Figure 15.** (**a**) Current waveforms without carrier shift control. (**b**) Current waveforms with carrier shift control. (**c**) FFT results of current waveforms.

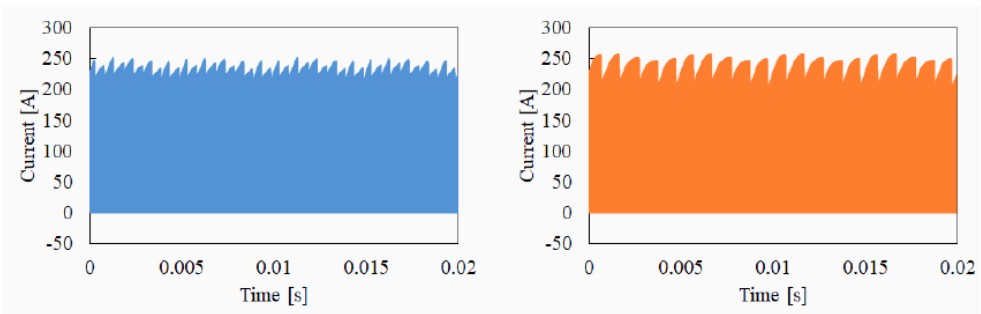

**Figure 16.** DC current ripple. (**Left**) shows the ripple without phase shift control and (**right**) shows it with phase shift control.

## 5. Experimental Results

This section shows some experimental results of the proposed method, winding changeover characteristics, 3rd harmonics current control, and PWM phase shift control. Figure 17 shows measured back EMF at no load condition, and Figure 18 shows measured torque characteristics. The measurement results are consistent with their FEA results. Then, the winding changeover technique was confirmed. Figure 19 shows the transient current waveforms when the mode is changed. Because the inductance energy is not terminated, the current and torque ripple is not generated when the mode is changed.

Figure 20 shows 3rd harmonics current control results and Figure 21 shows the output torque amplitude by changing the 3rd harmonics current phase. The current control shown in Figure 11 is satisfied, however, the output torque is not much increased compared with its FEA result even the change with the current phase is the same phenomenon because the 3rd magnetic flux is not large in the real machine.

Figure 22 shows effectiveness of the PWM carrier shift control. The current waveforms are much improved by the PWM carrier shift, high-frequency component that generates iron loss is decreased. Table 2 shows measured efficiency and loss in 1500 rpm 30 Arms condition. About 2.5% efficiency improvement is confirmed by the control. The effectiveness of its control is much better, especially in high-speed region.

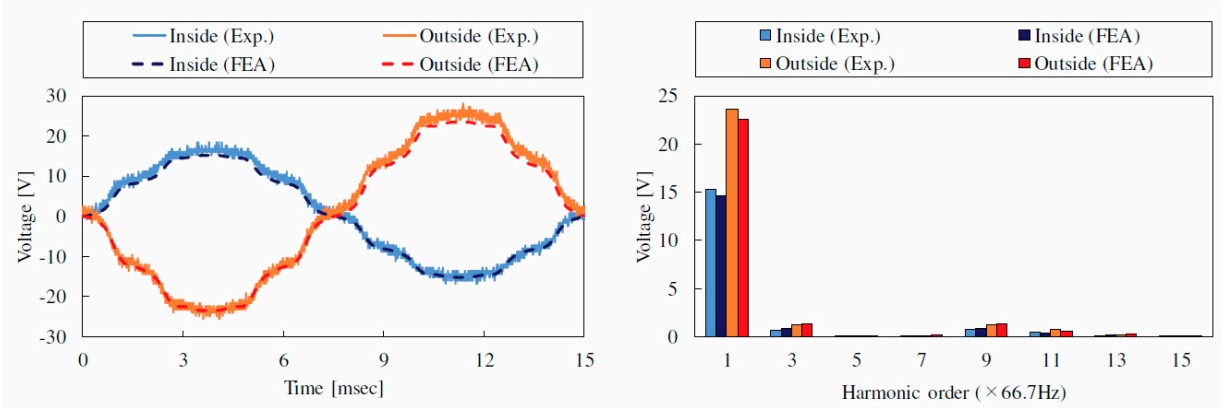

**Figure 17.** No load back EMF waveforms (**left**) and their FFT results (**right**) at 1000 rpm.

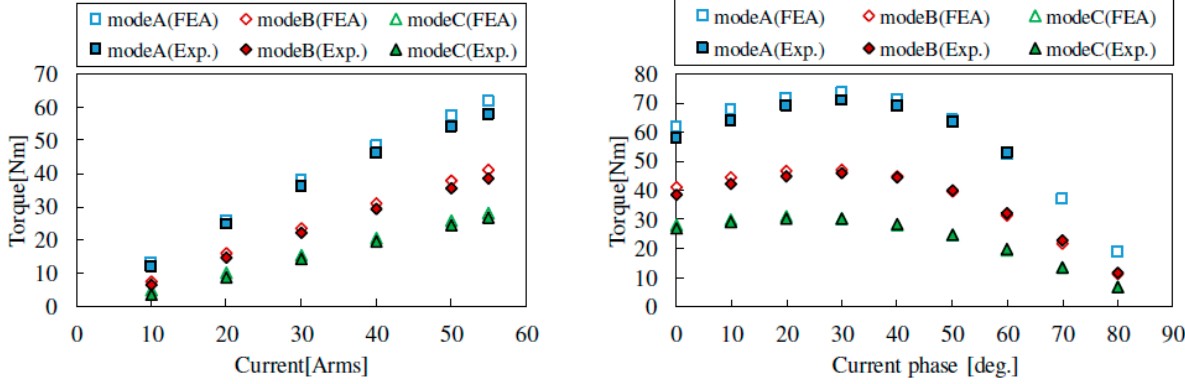

**Figure 18.** Measured and analyzed output torque in each mode. Left shows the current vs. torque and right shows the current phase vs. torque when current is 55 Arms.

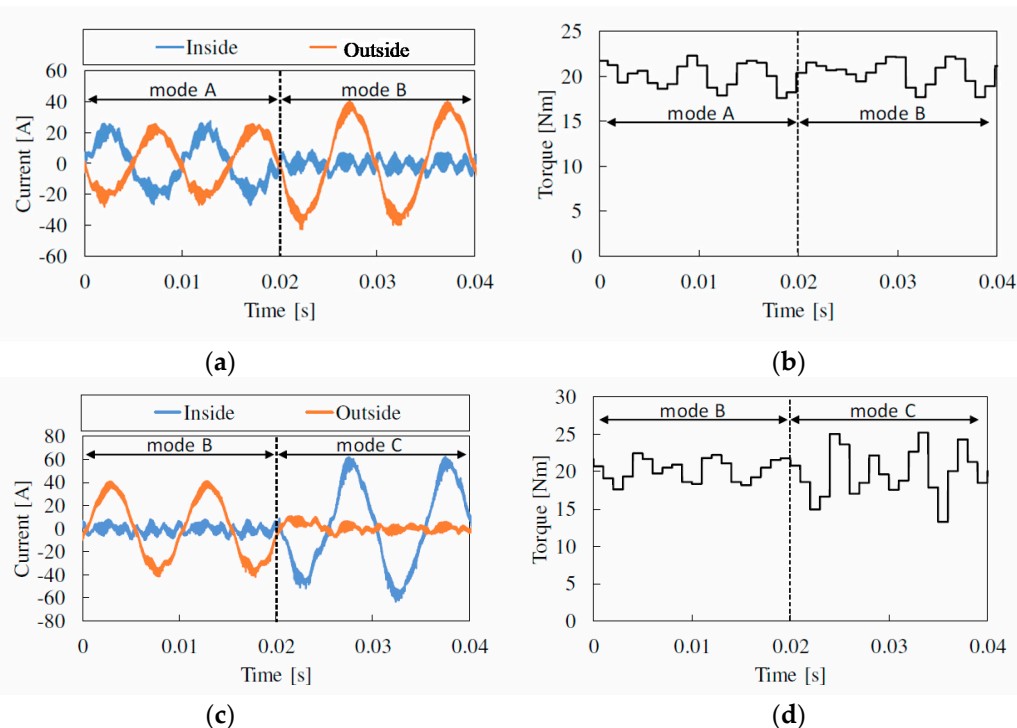

**Figure 19.** Experimental results of winding changing. Current waveform in each winding and torque waveform are shown. The current amplitude of each winding is calculated, the output torque is not to be changed. (**a**) Current waveform of each winding. The mode is changed from mode A to mode B. (**b**) Output torque waveform of each mode. The mode is changed from mode A to mode B. (**c**) Current waveform of each winding. The mode is changed from mode B to mode C. (**d**) Output torque waveform of each mode. The mode is changed from mode B to mode C.

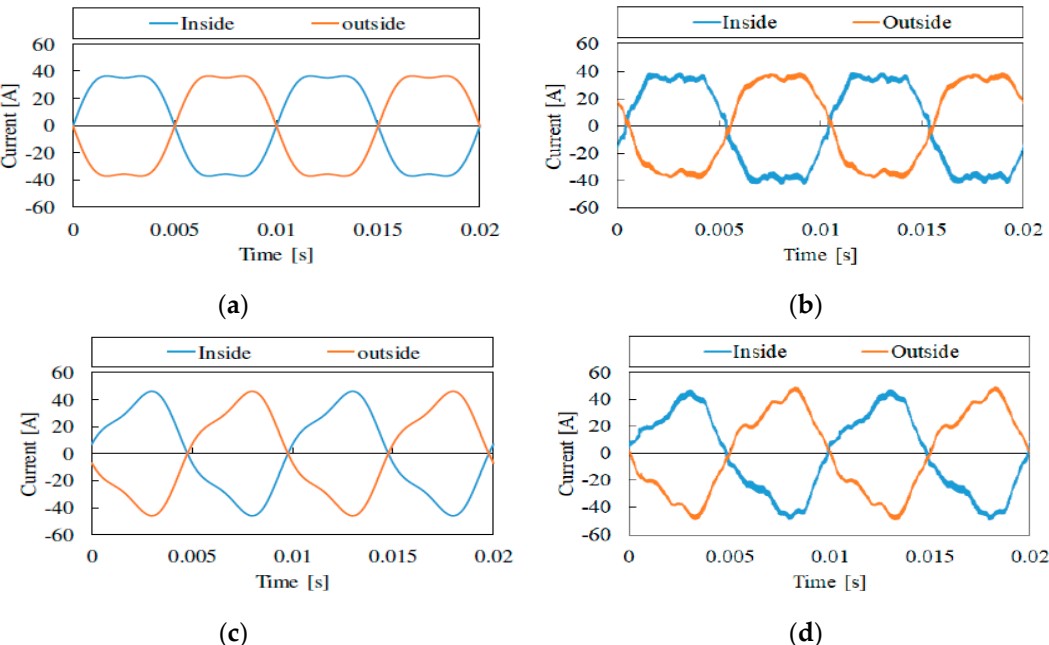

**Figure 20.** Measured current waveforms with 3rd harmonics current injection. Fundamental current amplitude is 30 Arms, 3rd harmonics current amplitude is 5 Arms. (**a**) Current reference of 3rd harmonics current injection, the phase is 0 deg. (**b**) Measured current of 3rd harmonics current injection, the phase is 0 deg. (**c**) Current reference of 3rd harmonics current injection, the phase is 90 deg. (**d**) Measured current of 3rd harmonics current injection, the phase is 90 deg.

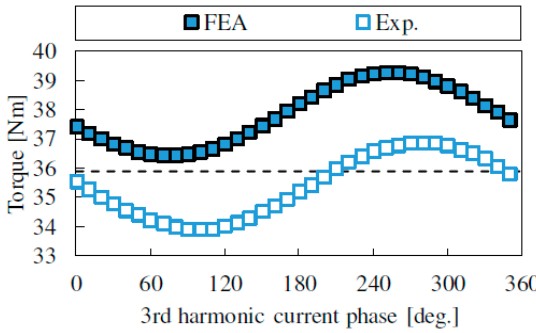

**Figure 21.** Output torque amplitude with 3rd harmonics.

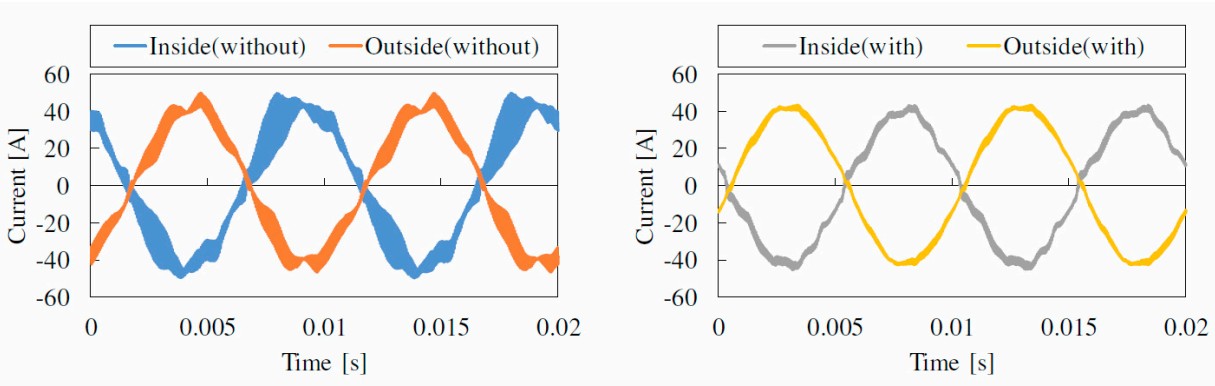

**Figure 22.** Measured current waveforms with/without PWM carrier shift control. (**Left**) shows it without control, (**right**) shows it with control.

**Table 2.** Measured efficiency and loss, 1500 rpm 30 Arms condition.

| Parameter | Without PWM Shift | With PWM Shift |
| --- | --- | --- |
| Efficiency [%] | 92.44 | 95.08 |
| Total loss [W] | 440.1 | 278.7 |
| Cupper loss [W] | 263.3 | 253.4 |
| Iron loss [W] | 176.8 | 25.3 |

## 6. Conclusions

This paper presented characteristics of the dual winding 5-phase PMSM for in-wheel motor, and evaluated unique control methods for the proposed dual winding 5-phase machine. The 5-phase motor is effective to realize fault-tolerant system, and the dual winding method can demonstrate winding change over and also realize additional fault-tolerant system for the in-wheel motor. The winding change over technique could increase efficiency in high-speed region because the reduced back EMF did not need the flux weakening current. The control methods, such as 3rd harmonics injection and PWM career shift control, were also demonstrated. The output torque can be increased by adding 3rd harmonics if 3rd harmonics flux is large enough in the motor. The iron loss reduction was achieved by PWM phase shift control by reducing PWM carrier harmonics current in the phase winding. The motor was installed in the 16-inch in-wheel motor, high-power density system was realized and performed.

**Author Contributions:** Conceptualization, K.A.; methodology, K.A. and K.F; software, K.F.; validation, K.A. and K.F.; formal analysis, K.F.; investigation, K.A.; resources, K.F.; data curation, K.F.; writing—original draft preparation, K.A.; writing—review and editing, K.A.; visualization, K.A.; supervision, K.A.; project administration, K.A.; funding acquisition, K.A. All authors have read and agreed to the published version of the manuscript.

**Funding:** This work was supported by Council for Science, Technology and Innovation(CSTI), Cross-ministerial Strategic Innovation Promotion Program (SIP), "Next-generation Power Electronics"(Funding agency:JST).

**Conflicts of Interest:** The authors declare no conflict of interest.

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
