# Peer review of "Advanced Control Method of 5-Phase Dual Concentrated Winding PMSM for Inverter Integrated In-Wheel Motor"

_wevj, doi:10.3390/wevj12020061_

Round 1
Reviewer 1 Report
This is an interesting article about the current control strategy in a 5-phase double star synchronous machine in wheel motor for a Nissan Leaf. Experiments are an asset. This is the continuation of several other articles that the authors have already written (ref 5, 6 and 8). However, the article seems to have been written too quickly. It deserves to review the text and add more references to the equations and previous results cited on which the work is based. The figures are too small and technical details on the realization of double layers winding are missing. Many typo errors must be fixed.
Comments:
Figure 3 needs references or please explain how to calculate it. What is your definition of the winding factor (probably fundamental magnitude vs max signal amplitude).
Figure 4 too small and there is an error on the a) (missing Z?). Please add references for this part.
Figure 5 is placed after the 6!
Variable geometry winding => may be explained the difference with Dalembert windings (multiple output windings for different turn number or pole configuration). Please describe deeper the necessary winding. It is necessary to have electrical isolated wingdings?
Part 3B needs more refs about previous work. Previous results are mentioned without refs or explanations. It is missing.
Double layers on teeth winding. Why not use the same wire with mid-length output? Need electrical separate coils? Chapter 3B can be rewritten: more refs or demonstration.
Figure 7: how to explain the curve (A) is falling to zero after 17500 rpm, while the other two B and C go there?
Effect of coil coupling not documented.
Figure 8: I am very surprised by the predominance of iron losses over Joules losses, error? Otherwise, need references!
The real topic of the article is beginning in chapter 4. Error on equation 2 of the Current vectors?
Last term of equation 5 with the mutual inductance have to be clarified.
Effect of magnetic saturation on performance? Effects with the variation of parameters for current control?
Improvement lead by harm 3? How do you define the phase of 3 harm? I guess Back EMF waveform is important regarding this effect of 3 harm? Back EMF must be showed ... it comes late.
Figures 19, square shape waveform for Torque? Effect of step by step simulation or low rate of acquisition system ?
Author Response
Thank you for your depth review. I attach the answer to all reviewers comments.

Reviewer 2 Report
I have following concerns about the proposed design:
- The authors should provide flux linkage results.
2. The PM machine have high cogging torque. Provide some analysis to the cogging torque.
3. In wheel machine have a problem of heating, which cooling technique is provided or should be provided?
4. Provide comparison to some existing design? How the proposed motor design is efficient as compared to existing design,
Author Response

(The authors gave the same response as above.)

Reviewer 3 Report
The paper brings valuable information and relevance. However, there are some critical points that must be solver from the authors.
Strongly suggest that papers change the title. It must reflect the real goal of the presented paper. As presented, it can be easily understood as reference [5].
Line 47 is also confuse since authors started the sentence with the reference. Its advised to use authors name and later present the reference. Author et. al. presents … [reference].
At line 72, is stated that the only wires that come into the motor are DC cables. However, there are no other cables such as control, or even measurement cables? At least, throttle signal must reach the motor. Strongly suggest to review.
Table I must state that output power is the max one, not rated.
There are a lot of problems with figures all over the paper. I’m not sure if there is a problem with the template, however, authors must pay attention: e.g. fig. 3 overlay page numbers, fig. 8 has poor quality, fig. 17 overlay the text “Figu”, and “000rpm”, fig. 19 legend: ourside.
The document must be better elaborated and reviewed before submission. If not, the presentation can be confused with the realized work. Please, improve paper quality.
There are also typos problems: line 190 eqaitions, fig. 19, line 380.
About presented results.
Its not clear if results presented in fig. 8 consider 3h injection.
Authors must present a discussion over 3h injection. For FEA evaluation, does it consider HB curve for behavior at 3h frequency? The steel performance change considerably with frequency increase..
Strongly suggest that authors review that section based upon:
Mengoni, Michele, et al. "High-torque-density control of multiphase induction motor drives operating over a wide speed range." IEEE Transactions on Industrial Electronics 62.2 (2014): 814-825.
How does the losses were evaluated?
Paper conclusion is poor. Which are the conclusions over operation modes? How does iron losses were evaluated? Authors state that torque is increased with 3h injection, however that is different from present in line 358. Maybe the text isn’t clear enough.
Author Response

(The authors gave the same response as above.)

Reviewer 4 Report
Good!
Author Response

(The authors gave the same response as above.)

Round 2
Reviewer 2 Report
The authors have presented a new design and control methods for five phase IM . I appreciate the authors work. I am satisfied with the author response. I have few minor comments :
The figure 4 (a) size should be matched with b, c, and d. The symbols used in equations in section 4 are not described. The authors should add future work for instance robust control techniques used in https://doi.org/10.3390/en13092158, 10.1109/INMIC50486.2020.9318199 for multiphase IM.
Best wishes
Reviewer 3 Report
paper quality has improved